# Larval Therapy and Larval Excretions/Secretions: A Potential Treatment for Biofilm in Chronic Wounds? A Systematic Review

**DOI:** 10.3390/microorganisms11020457

**Published:** 2023-02-11

**Authors:** Daniel Morris, Micah Flores, Llinos Harris, John Gammon, Yamni Nigam

**Affiliations:** 1Faculty of Medicine, Health, and Life Science, Swansea University, Swansea SA2 8PP, UK; 2BioMonde, Bridgend CF31 3BG, UK

**Keywords:** chronic wound, biofilm, larval therapy, wound healing, *Staphylococcus aureus*, *Pseudomonas aeruginosa*, *Lucilia sericata*, maggot therapy

## Abstract

Chronic wounds present a global healthcare challenge and are increasing in prevalence, with bacterial biofilms being the primary roadblock to healing in most cases. A systematic review of the to-date knowledge on larval therapy’s interaction with chronic-wound biofilm is presented here. The findings detail how larval therapy—the controlled application of necrophagous blowfly larvae—acts on biofilms produced by chronic-wound-relevant bacteria through their principle pharmacological mode of action: the secretion and excretion of biologically active substances into the wound bed. A total of 12 inclusion-criteria-meeting publications were identified following the application of a PRISMA-guided methodology for a systematic review. The findings of these publications were qualitatively analyzed to provide a summary of the prevailing understanding of larval therapy’s effects on bacterial biofilm. A further review assessed the quality of the existing evidence to identify knowledge gaps and suggest ways these may be bridged. In summary, larval therapy has a seemingly unarguable ability to inhibit and degrade bacterial biofilms associated with impaired wound healing. However, further research is needed to clarify and standardize the methodological approach in this area of investigation. Such research may lead to the clinical application of larval therapy or derivative treatments for the management of chronic-wound biofilms and improve patient healing outcomes at a time when alternative therapies are desperately needed.

## 1. Introduction

Chronic wounds present a complex and burdensome challenge to clinicians globally [1], with bacterial biofilms presenting a significant barrier to wound healing [2]. Effective treatment modalities and therapies that successfully target chronic-wound biofilms will expedite healing and improve patient outcomes, in addition to reducing the clinical and economic burdens in healthcare settings. Larval therapy (LT) is a well-studied and widely implemented chronic-wound treatment with a substantial history of clinical use [3,4]. This review aims to determine the extent and the quality of the evidence that exists regarding the potential of LT in biofilm management and its providing a reservoir of therapeutic compounds that hold promise for the development of novel biofilm treatments.

### 1.1. Chronic Wounds

Nonhealing or chronic wounds are wounds that do not progress through the recognized phases of timely reparative healing—haemostasis, inflammation, proliferation, and remodelling [5]—and are instead induced into a state of pathological inflammation [6]. Consequently, the disrupted healing process leads to poor anatomical and functional outcomes, the chronicity and frequent relapse of which carry risks of limb loss and major disability [7]. It is currently estimated that 1–2% of the population in developed countries will experience a chronic wound in their lifetime [8]. This, however, may be an underestimate, as chronic wounds are sometimes termed ‘a silent epidemic’, veiled as a comorbid condition of more distinct pathologies [9]. As such, they increasingly impact the health of the global population as life expectancy (the incidence of chronic wounding correlates with age) and diabetes incidence (which is correlated with common chronic-wound comorbidity) continue to trend upward [1,10,11]. Whilst the physiological sequelae and burdens on healthcare practitioners treating chronic wounds are clinically apparent, it is imperative to recognize the emotional, social, and psychological consequences of chronic wounding, too, which are often equally significant and longer lasting. Over 30% of patients with chronic wounds suffer from depressive symptoms or anxiety, which significantly reduces their quality of life (QoL) [12], further impairing healing given the established psychoimmunological relationship between stress and maladaptive wound-healing responses [13,14]. The health impacts aside, chronic wounds impose a significant economic burden, with costs estimated to be between 1 and 3% of total healthcare expenditure [15,16,17]. Again, this may be an underestimation, as more detailed analyses of Welsh National Healthcare Service healthcare expenditure in 2012–2013 revealed the cost of chronic wounds to be GBP 328.8 million per year, accounting for 5.5% of the total budget [16]. For the UK, the cost of treating chronic wounds was estimated to be GBP 4.5–5.1 billion per year [18], making it comparable to the management of a more widely recognized public-health crisis—obesity—which was determined to cost GBP 5.1 billion from 2006 to 2007 [19]. The majority (66%) of chronic-wound-associated costs are incurred in the community and secondary care, where nurses attend to, dress, and manage wounds [18]. More effective treatments that reduce the number of nurse interventions needed have the potential to significantly reduce the clinical and economic impacts of chronic wounding.

Several factors impair wound healing [20], including oxygenation [21], infection [22], age [23], sex hormones [24], stress [25], diabetes [26], obesity [27], medications [28], alcoholism [29], smoking [30], and nutrition [31]. Whilst individually impactful, typically, a multitude of these factors act in concert to disrupt the healing process in patients with chronic wounds. It has long been recognized that, following wounding and the consequent compromise of the protective barrier function of the skin, colonization and infection by microorganisms can impair healing and exacerbate inflammation [32]. Historically, the view was that the bacteria responsible for chronic infections behaved much the same as those grown in liquid culture media, so-called ‘planktonic’ bacteria. However, the past 30 years of research has resulted in a paradigm shift in the perspectives of microbiologists, with the current consensus being that most bacteria that exist in a natural or pathogenic state do so primarily as biofilms [33,34,35]. A biofilm is defined as an aggregation of localized microorganisms attached to a surface in a heterogenous community, housed within a fluid matrix of extracellular polymeric substances (EPS) [36]. EPS comprise a matrixome of exopolysaccharides, nucleic acids (eDNA and eRNA), proteins, lipids, and other biomolecules [37].

Bacteria initially attach to abiotic or biotic surfaces, with binding facilitated by various physiochemical and electrostatic interactions between the bacterial envelopes and the surfaces. Both specific and nonspecific mechanisms of attachment have been reported [38,39]. Abiotic surface attachment is the consequence of electrostatic interactions between a material and bacteria [40], whereas, for biotic surfaces, such as chronic-wound beds, attachment is regulated by specific receptor–ligand interactions [41]. Both the bacterial envelopes and the phospholipid bilayers of cells are net negatively charged, and thus electrostatic repulsion must be overcome for biotic surface attachment. To overcome this force, bacteria deploy specific organelles, such as flagella and pili, which propel or rappel to the cell surface [39]. Effective bacterial binding to surfaces induces cascades of complex intracellular signaling events that attenuate gene-expression patterns, shifting an attached organism’s phenotype from a planktonic to a biofilm state [42,43].

Biofilms have become increasingly associated with chronic wounds, with recent analyses revealing that ~80% of chronic wounds are colonized by biofilms [44]. Biofilms are difficult to treat because of their innate antibiotic-resistance properties, the levels of their resistance to antibiotics being 100–1000× those of their planktonic counterparts [45]. In addition to their inherent phenotypic recalcitrance to traditional antibiotics, biofilms provide environments that foster and fuel another major threat to public health: resistance-gene-determined antimicrobial resistance (AMR) [46]. This is achieved as biofilms enable exposure to subinhibitory concentrations of antibiotics coupled with high cell densities, increased genetic competence, and the accumulation of genetic elements or uptake of resistance genes [47]. Horizontal transfer of resistance genes via conjugation is the sole mechanism identified for biofilm-mediated resistance-gene acquisition, with several studies showing it to be more efficient in bacterial biofilms as compared with planktonic cells [48,49,50].

Directly targeting biofilms within chronic wounds has been shown to significantly improve wound-healing outcomes, with antibiofilm approaches utilized alongside standard of care working synergistically to improve healing [51]. As such, effective treatments and management practices addressing and resolving biofilm growth within wounds are much needed to mitigate the public health and economic impacts that chronic wounds have on societies globally. Universally, the first step in the treatment of chronic wounds is debridement [22,52,53,54,55,56]. Debridement is the process of removing nonviable tissue and organic matter, including biofilm, from the wound bed and can be achieved by several means: sharp, surgical, autolytic, enzymatic, mechanical, and larval therapy [55]. Whilst debridement is the first tool clinicians deploy in the treatment of chronic wounds and associated biofilms, like all wound-care applications, it must be employed as part of a structured wound-management plan [57]. It is well established that debridement alone, regardless of method, will not completely address the sequelae caused by chronic-wound biofilms, with biofilms rapidly re-establishing post-debridement [55,58,59,60]. Therefore, investigations into post-debridement infection management using existing debridement and chronic-wound treatment strategies are accelerating, with the focus shifting to addressing how they may be used to effectively manage biofilm within the wound bed [55,58,59,60,61,62].

### 1.2. Larval Therapy 

Larval therapy (LT), or, as it is otherwise known, maggot debridement therapy (MDT) or maggot therapy (MT), is a ‘bio-surgical’ procedure that involves the controlled application of medical-grade blow fly larvae (typically *Lucilia* spp.) to chronic and nonhealing wounds [63]. It was first introduced into modern medicine by the American orthopaedic surgeon William Baer for the treatment of osteomyelitis in the 1930s [4]. However, the anecdotal use of larvae in the treatment of wounds dates back centuries, with various ancient cultures, such as the Australian Aboriginal and Central American Mayan tribes, applying larvae to wounds to promote healing [3]. Though effective, the use of medicinal larvae declined significantly in the 1940s following the advent of antibiotics. However, due to the increasing threat of multidrug-resistant bacteria [64], which delay healing and reduce the effectiveness of therapeutics, interest in LT has gradually revived. Since its resurgence, LT has been steadily gaining traction in the clinical setting, leading to an ever-growing body of evidence supporting its efficacy, efficiency, and economy in the treatment of chronic wounds [65,66,67,68,69,70,71,72]. LT aids the healing of chronic wounds in three core areas: wound debridement [73], wound disinfection [74] and wound closure [75,76]. These wound-treating effects are results of the method by which the larvae feed in the wound, termed ‘extracorporeal digestion’. A necrophagous species, *L. sericata* larvae, excretes and secretes (excretions/secretions (ES)) a complex matrix of proteolytic, glycolytic, lipolytic, and nuclease enzymes that work symphonically to digest and degrade devitalized and necrotic tissue [77]. This enzymatic action is assisted by a suite of antimicrobial peptides, bioactive molecules, and chemical compounds that direct the chronic-wound environment from inflammatory stasis toward healing. It should be noted that, whilst effective, LT has its limitations and is often considered a last-resort treatment option [78], this being attributed to the ‘yuk/yuck’ factor [79,80]. The reluctance to use this therapy is common to both patients and clinicians; however, it has been reported that experienced clinicians educated in LT are more likely to prescribe the treatment, education increasing adoption [81]. Furthermore, live larvae have a limited shelf life and require stringent storage and transport conditions which serve as barriers to routine ‘off-the-shelf’ applications [82]. Therefore, isolating and characterizing the benefits of the active components of LT—the ES—for the development of new treatment strategies is highly desirable.

This review seeks to collate the existing clinical and pre-clinical evidence for the effect of LT on biofilms specifically to determine the extent of current knowledge, identify knowledge gaps, and decide whether further investigation is warranted.

## 2. Materials and Methods

A review protocol has not been published. However, this review was conducted in accordance with the Preferred Reporting Items for Systematic reviews and Meta-Analyses (PRISMA) Checklist [83].

### 2.1. Eligibility Criteria

Studies were only included in this review if they assessed LT or derivatives of larvae used for LT specifically against bacterial biofilms.

### 2.2. Information Sources

The bibliographic databases PubMed, Web of Science, and Cochrane Central were searched in March 2022.

### 2.3. Search

Search terms related to larval therapy and biofilms were used, combined with the Boolean operator ‘AND’. The search strategy used was as follows: (1) Larval Therapy AND Biofilm, (2) Maggot Therapy AND Biofilm, (3) Maggot AND Biofilm, and (4) *Lucilia sericata* AND Biofilm. Excretions/Secretions was also searched, but this yielded fewer and narrower results than the previously listed searches.

### 2.4. Selection of Sources of Evidence

Search results were imported into Microsoft Excel and duplicates were removed. Titles and abstracts were screened by a researcher, and those that did not fit the inclusion criteria were excluded. Potentially eligible full-text articles were screened by the researchers according to the inclusion criteria.

### 2.5. Data Charting Process

A data charting table was created in Microsoft Excel to determine which variables to extract. Data were extracted into the pre-defined fields below by the author.

### 2.6. Data Items

The following information was extracted: (a) Title, (b) Authors, (c) Year, (d) Extract, Collection Protocol, (e) Extract Quantification, (f) Instar of larvae, (g) Biofilm Assay, (h) Isolates, (i) Inoculum Quantification, (j) Confirmation of Biofilm Phenotype, (k) Media, (l) Biofilm Incubation Time, (m) Monomicrobial or Polymicrobial, (n) Control Type Used, (o) Biofilm Harvesting Method, (p) Biofilm Visualization, (q) Results Summary, (r) Mode of Action Determination. See Appendix A.

## 3. Results

### 3.1. Search Results

Based on the search terms listed in 2.3 Search, the initial search yielded 580 studies. Following the removal of duplicates, 362 articles remained. Through title and abstract screening, 327 articles were excluded as they failed to meet the inclusion criteria as listed in Section 2.1. Of the 35 remaining papers, 23 were excluded through full-text screening as they failed to meet the inclusion criteria (see PRISMA flowchart Figure 1 for details). A total of 12 articles were included in this review [82,83,84,85,86,87,88,89,90,91,92,93].

### 3.2. Characteristics of Evidence Sources

A summary table of the studies analyzed is presented below in Table 1. For the full data extraction table, see Appendix A. 

### 3.3. Synthesis of Results

Eighty-three percent (10/12) of the studies analyzed were in vitro assessments of the effects of larvae/larval secretions on biofilm [84,85,86,87,88,89,90,91,93,94], with 17% (2/12) of studies being ex vivo in nature, one using a porcine-skin-based model [92] and the other a human-skin-based model [95]. In 67% (8/12) of studies, a passive secretion collection strategy was utilized, whereby live larvae were immersed in a suitable diluent, such as sterile H_2_O or 0.9% phosphate-buffered saline (PBS) solution [84,85,86,87,88,89,90,93], whereas 17% (2/12) utilized an extraction process that involved the milling and further processing of dried *L. sericata* larvae [94,95], 8% (1/12) utilized a recombinant enzyme derived from *L. sericata* secretions [91], and 8% (1/12) applied live larvae directly to their biofilm model [92]. For those studies that did not use a larval-derived recombinant (11/12), 25% (3/12) did not state the instar of larvae used for their extraction or treatment protocol [89,92,94], 25% (3/12) employed 3rd-instar larvae [85,93,95], 17% (2/12) used instar 1 and 3 larvae [86,88], and 25% (3/12) collected secretions from instar 2 and 3 larvae [84,87,90]. Excluding the study that applied live larvae (1/12) [92], 73% of the remaining studies (8/11) used total protein concentrations to quantify their extracts [84,86,87,88,89,90,91,93], with 18% (2/11) quantifying their extracts in mg/mL (mg of milled larvae per mL of water) [94,95] and 9% (1/11) not quantifying their extracts [85]. Seventy-five percent (9/12) of the studies employed a 96-well-microtiter-plate-based biofilm assay [84,85,86,87,88,90,91,93,94], 8% (1/12) used a modified Lubbock chronic-wound pathogenic-biofilm model [89], 8% (1/12) used an ex vivo human-dermal-skin model [95], and 8% (1/12) utilized an ex vivo pig explant model [92]. The organisms used to generate biofilms in these studies were as follows: 67% (8/12) of studies used *Staphylococcus aureus* [84,87,88,91,92,93,94,95], 50% (6/12) used *Pseudomonas aeruginosa* [84,86,89,90,92,95], 25% (3/12) used *Staphylococcus epidermidis* [85,88,91], 17% (2/12) used *Enterobacter cloacae* [88,93], and *Klebsiella oxytoca* [88], *Enterococcus faecalis* [88], and *Proteus mirabilis* [93] were used in 8% (1/12) studies. Fifty-eight percent (7/12) of studies used wound- or contaminated-medical-device-derived isolates [85,88,89,90,91,93,95], with the remaining studies using ATCC or tissue-bank isolates. Forty-two percent (5/12) of studies quantified the bacterial inoculum used to generate biofilm: 3 used ~10^5^ CFU/mL [86,88,92], 1 used 1 × 10^6^ CFU/mL [89], and 1 used ~10^8^ CFU/mL [94]. Only two studies verified isolates that possessed a biofilm-forming phenotype pre-testing [85,86].

All studies reviewed applied larval secretions/extracts or live larvae to monomicrobial biofilm models; no polymicrobial biofilm platforms were tested. Furthermore, all experimental controls in all the studies reviewed employed untreated or no-treatment control groups to determine antibiofilm efficacy; no comparative or established antibiofilm compounds were used. Most studies—67% (8/12)—used tryptone soy broth (TSB) media to culture biofilms [84,85,87,88,90,91,93,94], and 38% (3/8) supplemented TSB with glucose to promote a biofilm phenotype [84,87,94]. Lysogeny broth (LB) was used in 8% (1/12) [89] and M63 broth was used for the culture of *P. aeruginosa* in 17% (2/12) of studies. The ex vivo biofilm models utilized antibiotic- and serum-supplemented DMEM [95] and pigskin on tryptone soy agar (TSA) [92], respectively. Additionally, *K. oxytoca*, *E. faecalis*, and *E. cloacae* were cultured in brain–heart infusion (BHI) broth in 8% (1/12) of studies [88]. Moreover, the ages of biofilms were also assessed, with 83% (10/12) of studies assessing the effect of larval secretions/extracts on 24 h biofilms. The remainder used 3-, 5-, 7-, and 9-day-old [88] and 3-day-old biofilms [92]. To determine the antibiofilm effects, researchers used optical densities (ODs) to quantitively determine the reductions in biofilm biomass pre- and post-treatment in 83% (10/12) of cases [84,85,86,87,88,89,90,91,93,94], and solubilization was used to harvest biofilms in all cases but one, which coupled solubilization with sonication [84]. The ex vivo pig explant model used CFU counts pre- and post-treatment to quantify biofilm growth [92], and the ex vivo human-skin model used the area of biofilm observed under light microscopy to determine the quantity of biofilm [95]. Visualization of biofilm was not performed in 42% (5/12) of the studies [84,86,87,88,93]. Scanning electron microscopy (SEM) was deployed in 33% (4/12) of studies to image biofilms [90,92,94,95], and light field microscopy was used in 33% (4/12) of studies [85,90,91,95], with fluorescence microscopy implemented in 2/12 cases [85,94]. Finally, transmission electron microscopy (TEM) was implemented in 1/12 cases [94], as was confocal microscopy [89]. All studies assessed in this review showed that larvae and/or their secretions had antibiofilm effects; all significantly reduced the formation of or degraded pre-formed biofilms. The concentration dependency of these antibiofilm effects was assessed in 25% (3/12) of studies [86,87,94]. For the degradation or inhibition of *S. aureus* biofilms, various results were reported; complete clearance 48 h post-treatment (5-log reduction in 24 h) for live larvae was reported for the pig explant model [92], with ~50% inhibition of formation and biofilm reduction in vitro with between 0.2 and 50µg/mL ES [84,87,88,93,94], whereas, for *P. aeruginosa*, in three separate studies, a minimum of ~20µg/mL ES was needed to observe a significant reduction in biofilm formation [84,86,89]—a quantity 10-fold higher than that required for similar antibiofilm effects to be observed in *S. aureus* [84]. Significant antibiofilm effects against *S. epidermidis* were reported in two studies [85,91], as were reported for *E. cloacae* [88,93], whereas significant reductions in biofilm formation with respect to *K. oxytoca* and *E. faecalis* were shown in isolation [88], *P. mirabilis* being the only organism tested for which no antibiofilm effect was shown [93]. In 33% of studies, researchers reporting antibiofilm effects did not endeavor to further elucidate the mechanisms underpinning their observations. One study concluded that larval extracts disrupted the permeability of *S. aureus* and *S. pneumonia* bacterial cell membranes, increasing their permeability [94]. For *P. aeruginosa*, van der Plas et al. inferred that the antibiofilm effects were not due to bacterial killing or quorum-sensing (QS) inhibition [84]. However, more recently, other researchers have shown that applying larval extracts to *P. aeruginosa* cultures decreased the expression of biofilm maturation and virulence genes (*lasR*, *rhlR*, and *rhlA*) [95] responsible for the Las and Rhl systems that regulate *P. aeruginosa* QS. Additionally, Brown et al. isolated a novel nuclease that digests components of *P. aeruginosa* biofilm, postulating that this enzyme is partly responsible for the observed effects of larval ES [89]. Furthermore, Harris et al. posited that enzymatic factors contained within larval ES degraded polysaccharide intercellular adhesin (PIA) and biofilm-associated accumulation-associated protein (Aap) in *S. epidermidis* through fluorescence microscopy observations of well-characterized strains post-treatment [85]. This was partially reconfirmed following the application of a recombinant larval chymotrypsin to the same strains of *S. epidermidis*, with antibiofilm activity observed for the Aap-dependent strain [91]. Bohova et al. sought to fractionate their larval ES by high-performance liquid chromatography (HPLC), finding that specific fractions that harbored the antibiofilm activity contained a protein with a molecular weight of 25 kDa [93]. Quantifying the time taken to inhibit or disrupt biofilms was assessed in 3/12 studies, with it being determined that 20–200 µg/mL eradicated *S. aureus* in 3 h [87], while for *S. epidermidis* it took 1–6 h (depending on the isolate) to disperse biofilms [85] and for *S. aureus* biofilm-formation inhibition took 8 h, compared with 10 h for *P. aeruginosa* [84]. Interestingly, Harris et al. showed the antibiofilm effects of larval ES to be temperature-dependent for one *S. epidermidis* isolate (no activity at 4 °C, moderate activity at 25 °C, and optimal activity at 37 °C) and temperature-independent (activity at all temperatures tested) for another *S. epidermidis* strain [85]. Furthermore, the heat stability of larval ES was explored in 3/12 studies; boiling ES reduced the efficacy but did not completely inhibit activity against *S. epidermidis* [85], while it abrogated the effects against *S. aureus*, but not against *P. aeruginosa* [84], and eliminated activity against *S. aureus* and *E. cloacae* [93].

## 4. Discussion

With an aging population and the rate of diabetes increasing globally, the ‘silent epidemic’ of chronic wounds, their resultant burdens on public health, healthcare costs, and contributions to AMR are only set to grow. Clinically addressing, managing, and effectively treating biofilms remains one of the largest unresolved barriers to healing patients with chronic wounds [96]. Therefore, chronic-wound-healing treatments must be developed for biofilms and incorporate strategies that target them [52]. Given the recalcitrance of chronic-wound biofilm to respond to traditional antibiotic therapies and the undesirable cytotoxic effects of currently employed antiseptics, the need for innovative new strategies to tackle chronic-wound biofilm is paramount. This may involve repurposing existing therapies, exploring the synergistic effects of multiple treatments, and developing new dressings, alginates, skin substitutes, preparations, and compounds [54]. Whilst the clinical effectiveness [97] and understanding of the costs and benefits [98] of LT continue to increase, alongside its antimicrobial effectiveness both in vitro [99] and in vivo [100,101], investigation of LT, larval secretions, and the molecules underpinning their effects against wound biofilms is pertinent, especially now that insects are being identified as underexploited reservoirs of therapeutic compounds [102,103,104]. The breadth of the literature investigating the antibiofilm effects of larval therapy specifically is narrow, with a total of only 12 papers, published between 2008 and 2022, recovered for assessment. However, all studies concluded that *L. sericata* larval secretions delivered significant and repeatable antibiofilm effects, warranting further exploration for their potential inclusion in antibiofilm treatments and protocols. The large degree of heterogeneity between the study protocols, however, makes meaningful comparisons between them difficult.

### 4.1. ES Collection Protocol

Most studies employed a passive-secretion collection protocol, where larvae were incubated in a suitable liquid medium. However, there exists a high degree of interprocess variability regarding the number of larvae, volume of diluent, diluent type, collection temperature, larval instar, collection time, post-collection processing (centrifugation, filtration, sterilization, etc.). Total protein concentration was employed in several instances post-collection to standardize and quantify ES; however, no further testing to quality control batches of generated extracts/secretions was listed. It has been determined that altering a single variable in the collection protocol alters the effect of ES on biofilm. For example, exposure to *P. aeruginosa* pre-collection promotes anti-*P. aeruginosa* biofilm activity in ES of equal protein concentration [90]. This suggests that the collection protocol significantly alters the biochemical constitution of ES, regardless of total protein concentration. Therefore, the normalization of *L. sericata* ES collection followed by more robust post-collection quality control (QC) would facilitate better characterization of larval ES, enhancing the reliability of results achieved interexperimentally. Pickles and Pritchard (2017) developed a semi-quantitative QC assay facilitating high throughput testing to verify serine protease activity in *L. sericata* secretions [105]; rapid and adaptable assays such as these could be used to verify the presence and concentration of key constituent molecules. As a natural product with a wide range of active constituent molecules, fully characterizing larval ES is a tall order. However, other topically applied natural wound-care treatments, such as Manuka honey, have well-defined biophysiochemical profiles that have led to improved ‘medical-grade’ honey (MGH), resulting in the clinical application of regulatorily accepted honey-based treatments [106,107,108,109]. As an insect-derived wound-care treatment, larval secretions could follow the roadmap laid out by MGH in its potential translation from benchtop to bedside.

### 4.2. Biofilm Models

Most of the reviewed studies were performed in vitro (10/12), with 2 ex vivo studies identified, displaying a distinct lack of in vivo research in this area, although this is not unique with respect to assessment of the efficacy of larval ES against biofilm, with 90% of testing for topically applied chronic-wound biofilm treatments being conducted in vitro [110]. Much like the heterogeneity in ES collection protocols, the methodological variation in the biofilm models reviewed makes the comparison of the overall efficacy of ES against biofilms between studies difficult. The capacity of in vitro models for rapid screening is a useful and essential tool in the early stages of optimizing and assessing the potential efficacy and safety of biofilm therapies [52]. As such, a sound body of in vitro evidence is often prerequisite before moving to animal and in vivo models [111]. Nevertheless, as with any testing system, it is only valuable if it has low user and situational variability and if in vivo conditions are effectively replicated [112]. Amongst the studies reviewed, all the in vitro models grew biofilms on abiotic surfaces, such as plastic and steel. Whilst useful for screening the efficacy against biofilms on implantable devices or other medically relevant materials [113], these models do not represent the microbial–host tissue interactions of biofilm within a chronic wound [114]. Biofilm-related chronic-wound infections in vivo grow on the surface of, or are suspended within, the quasi-solid matrices of the wound bed [115]. The necrotic tissue, slough, and wound exudate present in chronic wounds consist largely of collagen [57], providing attachment sites for pathogenic bacteria and subsequent biofilm formation [116]. Thus, collagen-based gel matrices have been used to cultivate biofilms in vitro to better mimic the chronic-wound environment [117,118,119,120]. With debridement as the primary antibiofilm treatment method [55,121], the established debridement efficacy of LT [97,122], the isolation of collagenases in *L. sericata* ES [123,124,125,126], and the therapeutic benefits of collagenase-based wound treatments [127], further study of the effects of ES on biofilms grown in collagen-based wound-simulating models may help to elucidate and characterize the interplay between ES constituent molecules and biofilms, providing more clinically relevant results and insights.

### 4.3. Bacteria

In the studies reviewed here, the primary bacterial species with respect to which larval ES was assessed were *S. aureus* and *P. aeruginosa*, these being the bacteria most associated with chronic-wound infections [128]. According to these studies, the efficacy of larval secretions in disrupting the biofilms of these species and their efficacy against other wound-relevant pathogens, such as *S. epidermidis* and *E. faecalis*, warrant further investigation. However, continuing with the theme of translational relevance between in vitro investigations and in vivo applications, all the studies—even if they involved screening against several bacteria—assessed the application of larval ES to monomicrobial biofilms, whereas chronic-wound biofilms are often polymicrobial [129]. The pathogeneses of polymicrobial infections differ significantly from those of monomicrobial diseases, with enhanced pathogen persistence at the infection site, increased severity, and greater recalcitrance to antimicrobial treatments [130]. So significantly do they differ, that it is accepted that Koch’s postulates need not be applied to wound infections [130]. This disparity in behavior may partly explain the decline in efficacy observed when treatments tested on monomicrobial in vitro models are submitted to in vivo testing [110]. Equally, the reverse may be true, in that potential treatments that disrupt the synergy between organisms within a polymicrobial biofilm are overlooked due to poor performance in monomicrobial models. This being so, several polymicrobial wound-biofilm in vitro assays have been developed that better recreate the chronic-wound environment [131,132,133]. Testing larval ES utilizing one of these models may provide greater insights into the efficacy of ES in the treatment of chronic-wound biofilm, strengthening or diminishing its applicability in vivo.

### 4.4. Media

Another environmental factor shown to affect biofilm formation and the subsequent efficacy of antimicrobial treatment is the media in which biofilms are generated [134,135]. All the in vitro studies of larval ES used a variation of standard laboratory culture media—either TSB [84,85,87,88,90,91,93,94], lysogeny broth (LB) [89], or M63 [84,86]. Whilst these media support the growth and development of biofilms in vitro, their nutrient densities and compositions do not simulate the in vivo situation well, with bacteria cultured in TSB being shown to have increased resistance to clinically employed antibiotic treatments [134]. This issue has become increasingly recognized, with several researchers developing chronic-wound-fluid-mimicking media for in vitro biofilm treatment screening [135,136,137]. Wound-bed fluid has been characterized as very serum-like in its composition [138,139] and using serum-based media that incorporate other important host-derived factors, such as fibrinogen, collagen, and fibronectin, have been shown to recreate the in vivo biofilm characteristics of 3D structure, biomass, metabolic activity, and polymicrobial coexistence when biofilms are cultured in vitro [137].

### 4.5. Inocula

There also existed a large degree of heterogeneity in the bacterial inocula applied in the ES/LT-treated biofilm models reviewed, with most studies (7/12) not quantifying the inoculum used and instead opting to implement log-phase cultures of undetermined concentrations [80,81,84,85,85,87,93]. The remainder of the studies used bacterial inocula of 10^5^ to 10^8^ CFU/mL [79,86,86,92,94]. Whilst quantifying starting inocula would facilitate interstudy comparisons, the clinical relevance is questionable given that there is no reference threshold that serves as an indication to inform clinicians of wound bioburden. Several studies have inferred that >10^4^ CFU/g is indicative of a pathogenic level of wound bioburden that impairs healing [140,141,142]. However, this approach is often thwarted by the variation in interspecies virulence; not all levels of bioburden are equal, with highly virulent strains of pathogenic bacteria requiring fewer CFUs than other species to cause wound infection and delay healing. As such, when tested, many wound-bioburden quantification techniques employed to inform clinical practice are found to be unreliable and can often do more harm than good [143]. Recommending a standardized approach to quantifying wound-relevant bacterial inocula that harmonizes research and clinical outcomes by translating in vitro data into clinical efficacy is beyond the scope of this review. However, if LT and its derivatives’ effects on wound biofilms are to be better understood and established, it would be prudent to quantify test inocula to >10^4^ CFU/mL to ensure their consideration by researchers and clinicians.

### 4.6. Biofilm Age

Another factor shown to determine biofilms’ susceptibility to treatment is their age/maturity [134,144,145,146]. It has been noted that as biofilms mature their tolerance to treatment increases [146], with Wolcott et al. noting that biofilms were most susceptible to treatments within the first 24 h of their formation [144]. In agreement with this, Phillips et al. determined that *P. aeruginosa* and *S. aureus* cultures required a minimum of 72 h to form biofilm structures that were tolerant to traditionally employed antimicrobials [147]. Of the 12 studies reviewed, 10 assessed the efficacy of *L. sericata* ES against 24 h biofilms of various species, including *P. aeruginosa* and *S. aureus*. As this is when these biofilms are supposed to be most susceptible to treatment, repeating these studies on mature biofilms (>72 h in age) would be of value. The ex vivo porcine biofilm model used by Cowan et al. suggests that the efficacy of LT is maintained against mature biofilms, demonstrating complete biofilm eradication of 72 h *P. aeruginosa* and *S. aureus* biofilms following the application of live larvae to the biofilm models [92]. Whether these effects are maintained when the physical larvae are removed from the equation and just ES is applied remains to be seen.

### 4.7. Treatment Time

Additionally, another set of juxtapositions that became apparent when reviewing the in vitro assessment of LT and *L. sericata* ES against biofilms, in contrast to standard-of-care clinical practice, were the disparities in application and treatment times. For example, all studies included in this review implemented a single application of LT or *L. sericata* derivatives to biofilm models and determined their impacts, whereas, in practice, most treatments require several to tens of applications. For example, heavily exudating chronic wounds can have dressings replaced one to three times daily [148]. When LT is employed, patients can often receive two or more successive dressing applications [79]. Given this, it would be worth determining whether successive applications of LT or ES in vitro have a greater impact versus single applications and whether any effects are compounded over time. Ultimately, complete eradication of chronic-wound biofilm is the goal of any antibiofilm treatment. Quantifying the number of LT/ES application(s) required to eradicate biofilm in vitro could also inform clinical practice and set expectations as to suspected treatment durations.

### 4.8. Vector

Moreover, if *L. sericata* ES or derivatives thereof are to be developed into clinically employable treatments and considered in a new wave of biofilm therapies, vectors that effectively administer them into wound beds are necessary [149]. The studies reviewed all ES, extracts, or live larvae directly applied to biofilms in vitro, which, whilst useful for inferring foundational antibiofilm effects, provided only a limited view of their clinical potential. It has previously been shown that *L. sericata* ES can be delivered through hydrogels to promote wound healing in vitro [150]. Similarly, recombinant *L. sericata* chymotrypsin has also been delivered via hydrogel, displaying debridement-relevant properties ex vivo [151]. Determining whether the antibiofilm effects of ES are likewise maintained when incorporated into commonly employed chronic-wound-care biomaterials would bolster the proposition of their use and their clinical translatability.

### 4.9. Mode of Action

Most studies reviewed sought to further elucidate the mechanisms of action that underpinned the observed antibiofilm effects of *L. sericata* ES. Reviewing the existing literature in its totality, there was found to be not a single mechanism responsible, but rather a host of factors which act on a multitude of cellular and biochemical pathways to inhibit the formation of and degrade biofilms. As the genetic mechanisms that underlie biofilm formation are increasingly understood for commonly encountered chronic-wound pathogens, such as *S. aureus* [152,153] and *P. aeruginosa* [154,155], determining the effects of therapeutic compounds on biofilm-associated gene expression offers a high-throughput technique that may provide further insights into the mechanisms underpinning experimental observations. As such, antibiofilm strategies—involving *L. sericata* ES or otherwise—may be refined and enhanced as the resolution of the interplay between compounds, gene expression, and biofilm formation increases. Becerikli et al. showed that *L. sericata* extract decreased the expression of biofilm-maturation and virulence genes in *P. aeruginosa* [95]. Repeating similar studies on other chronic-wound-relevant biofilm-forming strains and species may offer views as to which species-specific biofilm-related genetic mechanisms are impacted by the application of *L. sericata* ES. Moreover, further fractionation and characterization of the molecules responsible for any biofilm-associated gene-expression modulation could elucidate molecular and cell–receptor interactions with therapeutic potential. In addition to exploring the influence that ES has on the gene expression profiles of bacterial biofilms, of equal validity is the exploration of the effects of bacterial biofilm on larval gene expression and subsequent secretion composition. To illustrate this, McKenna et al., 2022, demonstrated the differential immune-related gene-expression patterns in larvae exposed to wound-relevant pathogens [156]. Determining how these translate into augmented secretion composition and how such modulations translate to antibiofilm efficacy may help resolve the molecular interactions that are responsible for the observed antibiofilm effects of ES.

## 5. Conclusions

Chronic wounds are a snowballing threat to public health globally [157]. They are widespread, clinically intensive, expensive, and hard-to-resolve pathologies, the colonization of which by bacterial biofilms exacerbates the healing process [158]. Effective antibiofilm treatment strategies are desperately needed [54], and there are a plethora of novel strategies and compounds in development to address this pressing issue [158,159,160]. Many workers seek to co-opt, enhance, and reinvigorate existing wound-care treatments [59,161], whilst others look to innovate new treatments, such as antimicrobial peptides (AMPs) [162,163], topical antiseptics [164], nanotechnologies [165,166], or photodynamic therapy [167,168], to target chronic-wound biofilm. Larval therapy is an established and efficacious chronic-wound debridement therapy, achieved through the extracorporeal digestion of devitalized tissue by *L. sericata* ES. There is a growing body of literature characterizing the beyond-debridement wound-healing properties of *L. sericata* ES and its constituent molecules. This review sought to collate the existing evidence on the antibiofilm properties of *L. sericata* ES to determine whether their consideration in the development of new clinical treatments is worthwhile. Whilst limited, the data indicate that *L. sericata* ES and derivatives thereof are effective against the bacterial biofilms formed by chronic-wound pathogens. As such, elementary research in this area has laid a foundation upon which more refined and directed research can be built. Studies that more closely explore the in vivo situation will serve to optimize LT/ES treatment protocol options and ensure translatability. High-throughput bacterial transcriptome analysis assays could aid in elucidating the underlying mechanisms responsible for the observed antibiofilm effects. Further exploring and characterizing *L. sericata* ES in the context of antibiofilm treatments while simultaneously considering clinical applicability is recommended.

## Figures and Tables

**Figure 1 microorganisms-11-00457-f001:**
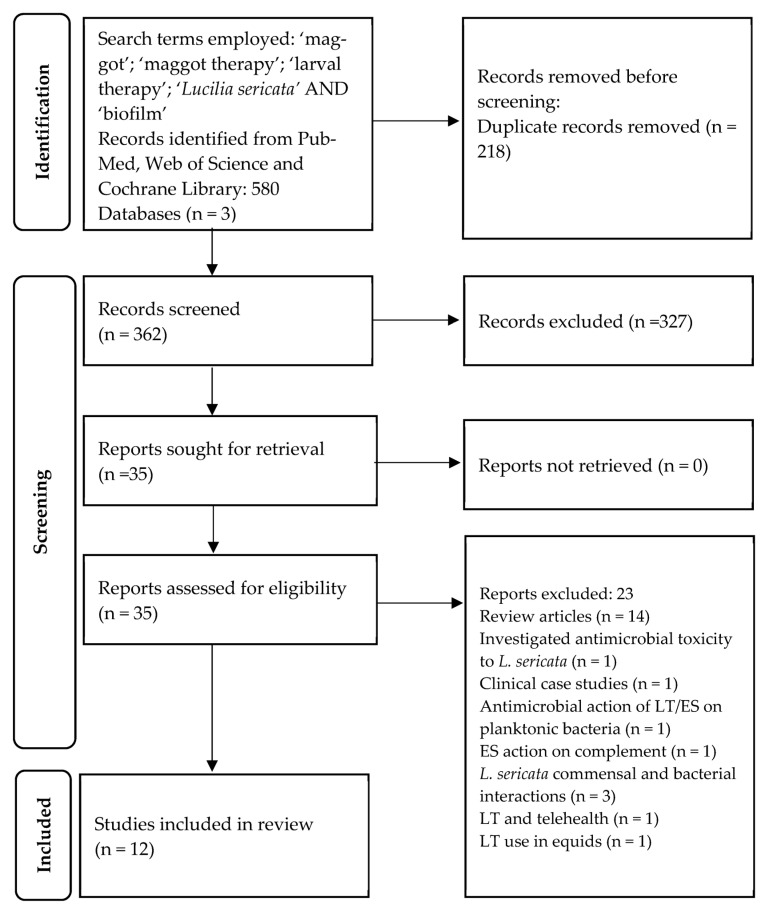
A PRISMA flowchart detailing the sources of literature and studies retrieved, reviewed, assessed for eligibility, and analyzed as part of this review.

**Table 1 microorganisms-11-00457-t001:** Summary of papers analyzed following the PRISMA-informed literature search.

Title	Test Agent	Biofilms	Results Summary	Reference
Maggot excretions/secretions are differentially effective against biofilms of *Staphylococcus aureus* and *Pseudomonas aeruginosa*	ES	*S. aureus* *P. aeruginosa*	A quantity of 0.2 µg of ES abolished *S. aureus* biofilm formation: 8 h incubation. Degradation of *P. aeruginosa* biofilms > 10 h incubation; required 10-fold more ES than *S. aureus* biofilms. Boiling of ES abrogated their effects on *S. aureus* but not on *P. aeruginosa* biofilms.	[84]
Disruption of *Staphylococcus epidermidis* biofilms by medicinal maggot *Lucilia sericata* excretions/secretions	ES	*S. epidermidis*	In the presence of ES, *S. epidermidis* 1457 and 5179-R1 nascent biofilm formation was inhibited, and pre-formed biofilms were disrupted. ES activity was temperature- and time-dependent, inactivated by heat treatment, and disruption depended on the mechanism of intercellular adhesion.	[85]
The Influence of Maggot Excretions on PAO1 Biofilm Formation on Different Biomaterials	ES	*P. aeruginosa*	Maggot ES prevents and inhibits PAO1 biofilm formation and degrades existing biofilms. ES still had considerable biofilm-reduction properties after storage at room temperature for 1 month. ES from instar-3 maggots were more effective than ES from instar-1 maggots.	[86]
Combinations of maggot excretions/secretions and antibiotics are effective against *Staphylococcus aureus* biofilms and the bacteria derived therefrom	ES	*S. aureus*	A quantity of 20–200 mg/L ES eradicated *S. aureus* biofilms within 3 h. Enhanced antimicrobial activity of daptomycin against biofilms.	[87]
Maggot excretions inhibit biofilm formation on biomaterials	ES	*S. aureus* *S. epidermidis* *K. oxytoca* *E. faecalis* *E. cloacae*	The presence of excretions/secretions reduced biofilm formation on all biomaterials. A maximum of 92% of biofilm reduction was measured.	[88]
Blow fly *Lucilia sericata* nuclease digests DNA associated with wound slough/eschar and with *Pseudomonas aeruginosa* biofilm	ES	*P. aeruginosa*	A quantity of 20 µg/mL ES resulted in an ~50% reduction in pre-formed biofilms.	[89]
Excretions/secretions from bacteria-pretreated maggot are more effective against *Pseudomonas aeruginosa* biofilms	ES	*P. aeruginosa*	Researchers stated that ES obtained from larvae pre-treated with 1 × 10^6^ CFU/mL *P. aeruginosa* displayed enhanced inhibition of nascent biofilm formation.	[90]
*Lucilia sericata* chymotrypsin disrupts protein adhesin-mediated staphylococcal biofilm formation	Recombinant larval derived enzyme	*S. aureus* *S. epidermidis*	Chymotrypsin derived from maggot excretions/secretions disrupts protein-dependent bacterial biofilm-formation mechanisms.	[91]
Chronic Wounds, Biofilms and Use of Medicinal Larvae	*L. sericata* larvae	*S. aureus* *P. aeruginosa*	Biofilms of *P. aeruginosa* and *S. aureus* grown on dermal pig explants were eradicated (6-log reduction) following a 48-h application of live *L. sericata* larvae. Following 24 h exposure, a 5-log reduction was observed.	[92]
Selective Antibiofilm Effects of *Lucilia sericata* Larvae Secretions/Excretions against Wound Pathogens	ES	*S. aureus* *E. cloacae* *P. mirabilis*	Maggot ES at 100 mg/mL concentration significantly reduced biofilm formation and disrupted established biofilm of *E. cloacae*. Heat-treated ES did not show any antibiofilm activity towards *E. cloacae*. Similar results were obtained in the case of *S. aureus*; however, the heat-treatment of maggot ES did not affect its antibiofilm activity.	[93]
Antibacterial and antibiofilm effects of fatty acids extract of dried *Lucilia sericata* larvae against *Staphylococcus aureus* and *Streptococcus pneumoniae* in vitro	Fatty acid extraction from dried and crushed *L. sericata* larvae	*S. aureus* *S. pneumoniae*	The fatty acid extract successfully inhibited the formation of biofilm and degraded mature biofilm produced by both species tested. Antibiofilm effects were concentration dependent.	[94]
Maggot Extract Interrupts Bacterial Biofilm Formation and Maturation in Combination with Antibiotics by Reducing the Expression of Virulence Genes	*L. sericata* extract	*S. aureus* *P. aeruginosa*	Significant reduction in observed biofilms in ex vivo human-dermal-skin explant model for both bacteria treated.	[95]

## Data Availability

Not applicable.

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
