# Peer review of "Larval Therapy and Larval Excretions/Secretions: A Potential Treatment for Biofilm in Chronic Wounds? A Systematic Review"

_microorganisms, 2023, doi:10.3390/microorganisms11020457_

Round 1
Reviewer 1 Report
Dear Authors, congratulations for the paper
Author Response
Many thanks to reviewer 1, for taking the time to read and consider this manuscript. There are no points to address, but an updated manuscript is available, should reviewer 1 choose to re-review the work.
Reviewer 2 Report
Biofilm-induced chronic wound infections are among the most significant causes of treatment failures. Healing chronic wounds with biofilm is a major issue worldwide due to high costs and adverse effects on patients.
This review intended to collate the clinical evidence for the effect of Larval Therapy on biofilm, and extent of current knowledge.
Corresponding information could be useful in the therapists performing maggot therapy and the management of chronic wound biofilms.
As a result, I consider that this work can be useful for the scientific community and the therapists performing maggot therapy as well.
However, some improvement needs to be done.
Specific suggestions for the Authors.
Please take most of these as suggestions for readability.
Line 15: please change “PRSIMA-guided” with “PRISMA-guided”
Line 69: delete (…wound healing) “2”
Line 74: i don't diggest "sterile barrier function of the skin" in line 74. please expressing it as "... compromise of the protective barrier function of the skin" instead of "... compromise of the sterile barrier function of the skin"
Line 177: please change “reviewer” with “researchers”
Author Response
Many thanks to reviewer 2 for taking the time to review and consider this papers suitability for publication and offering revisions that improve the quality of the paper. All readability issues highlighted have been amended as per the reviewers request.
Reviewer 3 Report
I have read your work with attention. I am very impressed with the precision and presentation of the problem. This review may change the thinking of clinicians who still believe that MDT is an alternative adjuvant method.
The introduction is well thought out and clearly introduces the issues, please consider whether to use the MDT terminology as well (maggot debridement therapy is more often used in the literature than larval therapy and in my opinion sounds more professional. The methodological and result part is understandable and legible, it does not raise my Please include in the discussion a few sentences of the text regarding the perception of MDT in the group of nurses in terms of implementing the method by medical personnel in everyday practice, as the presented results clearly indicate antibacterial and antibiofilm activities. Congratulations again and thank you for preparing this work. Prof. D bazaliński
Author Response
I thank reviewer 3 for taking the time to review and offer a verdict as to the suitability of this manuscript for publication and for the suggested improvements. The inclusion of the recommended paper on the readiness of nurses to undertake MDT as part of their practice has been included, as suggested, in the introductory segment to highlight the need for increasing institutional awareness of MDT as an effective therapy. The use of which is often inhibited by a lack of education. Many thanks for the review and subsequent comments.